# De Novo Shoot Development of Tropical Plants: New Insights for *Syngonium podophyllum* Schott.

Camelia Sava Sand and Maria-Mihaela Antofie *

Faculty of Agricultural Sciences, Food Engineering and Environment Protection, Lucian Blaga University of Sibiu, 7-9 Dr. Ioan Ratiu, 550012 Sibiu, Romania
* Correspondence: mihaela.antofie@ulbsibiu.ro

**Abstract:** *Syngonium podophyllum* Schott. cv. 'White Butterfly' is recognized as a valuable ornamental plant, and today it is also an important plant species of medicinal interest due to its high contents of phenolic compounds. The purpose of this article is to review the main scientific publications from our laboratory with regard to new scientific achievements dealing with *Syngonium* species or topics of interest, such as callus formation and further de novo shoot regeneration. The principles and stages necessary to start an industrial-level micropropagation protocol are discussed based on our experience. Different media compositions induced different morphogenetic responses inside the callus—particularly those related to the development of xylematic elements in the organogenetic areas, such as those for rooting, protocorms, and de novo shoot formation. The re-evaluation of old histological images revealed for the first time that xylematic elements are constantly closely positioned to all organogenetic centers, and that their development is closely dependent on the composition of the culture medium. Separate protocorms can be identified only when xylematic tracheary elements are well developed and closely connected to them. The formation of protocorms is strongly dependent on the mineral composition of the culture medium and the balance of plant growth regulators.

**Keywords:** *Syngonium*; callus; shoots; minerals; cysteine; xylem; protocorms





## 1. Introduction

The organogenesis of shoots is an essential developmental step in the more complex process of plants' morphogenesis for in vitro micropropagation [1]. More than a century ago, Gottlieb Haberlandt stated that plant cells may undergo totipotency—a complex physiological process that, under appropriate conditions, implies somatic plant cells expressing their full genetic information and having the ability to develop into a whole new plant [2]. In 1935, George Avery defined the auxin concentration gradient in plants as being expressed much higher from the tip to the base of the leaf; he also mentioned the 1928 discoveries by Frits Warmolt Went regarding the quantitative determination of auxins using oat seedlings [3]. This discovery was considered to be pivotal in his working group [4] and was followed by a series of other discoveries related to plant growth regulators, which were later underlined by Folke Skoog in 1951 [5].

We should also highlight the contribution of Philip White using tobacco callus tissue culture and, furthermore, his scientific success in obtaining fully developed de novo shoots [6]. The year 1938 can be considered to have brought new scientific evidence proving that an apparently undifferentiated tissue, under controlled conditions, may undergo some differentiation processes that will eventually produce tobacco buds. The principle of totipotency was confirmed by Philip White in 1934 [7]. Moreover, it took only a few steps to prove over time that more and more plant species had become subjects for study due to this ability to produce callus and furthermore to produce the entire plant body from callus tissues.

In the meantime, scientists have raised questions such as the following: What types of donor plant tissues are most suitable to be used for in vitro cultures? How are shoots formed from callus tissues? What processes take place inside the callus? Why do some areas undergo de novo shoot development while others do not? [8–10]. Later, in 1970, based on hundreds of scientific results as well as their own experience, Frederick Steward and collaborators stated that all somatic plant cells can potentially express totipotency under appropriate conditions [11].

Attila Feher recently published a relevant scientific article related to the review of the use of terms in this scientific field [12]. While callus was defined as a growing mass of non-organized cells in 1957 [13], Attila Feher presented new insights related to the need to redefine the structure and organization of callus by taking into account molecular makers already described in *Arabidopsis thaliana* for the processes of dedifferentiation, differentiation, and transdifferentiation. Feher agrees that callus should no longer be considered to be an unorganized tissue but, rather, a result of transdifferentiation processes, further supporting the 2011 observations of Kaoru Sugimoto, Sean Gordon, and Elliot Meyerowitz [14]. Based on their scientific results, auxin-induced callus formation appears to express lateral roots markers instead. Moreover, there seem to be relevant differences between different types of calli, which originate from different plant tissues according to Attila Feher [12]. In this regard, it is also essential to mention the range in the levels of plant growth regulators used to induce callogenesis, as well as to further scientifically substantiate the understanding of organogenetic processes.

Kaoru Sugimoto and collaborators also stated that callus is rather heterogeneous in its composition, being formed of different cell types, and only a few callus cells may ultimately be able to undergo organogenesis and somatic embryogenesis [14]. In this regard, Attila Feher considers callus to be the result of cell reprograming due to environmental conditions, and a single cell can become totipotent if that cell forms a completely new plant via somatic embryogenesis [12]. Based on these scientific findings, the idea arises that callus is not a totipotent or pluripotent tissue [14] but, rather, an intermediary living structure developed under stress factors, which is not controlled by the plant itself, acting as a new independent body that can contain different types of cells, only some of which can become totipotent or pluripotent, being involved in regenerating the whole plant body or only certain organs [12].

The analysis of the main genetic factors supporting de novo shoot formation has been of great interest among scientists [15]. Today, it is well established that auxin-induced de novo shoot formation may originate from perivascular cells, similar to the xylem-pole pericycle cells that initiate the development of lateral roots in *A. thaliana* [16]. These authors also define the concept of a "small founder group of cells" involved in shoot organogenesis. Furthermore, they mention that in order to fully understand the molecular mechanisms, it is necessary to apply synergic cytological, molecular, and genomic approaches [16].

Later, other authors concluded that de novo organogenesis is a complex process that takes place via three logical stages: (1) the activation of regenerating cells, (2) the acquisition of competency, and (3) initial de novo establishment of the apical meristem that can stimulate shoot formation [17].

Today, based on accumulated scientific information on the subject, the genetic pathway for shoot regeneration without plant growth regulators—by altering the expression levels of a group of several genes controlling the process—has already been patented [18].

In direct connection with the process of callus formation at the sites of plants' wounds, abundant relevant scientific evidence has been accumulated. Thus, it is well established that wounded tissues or organs are able to form callus to seal the wounds, which may indicate either adventitious organogenesis or somatic embryogenesis, based on distinct developmental pathways that require specific decisions on the cell fate [19]. These authors suggest that organogenesis at the callus level depends on the balance between three factors: (1) the components of the lipid transfer pathway, (2) reactive oxygen species (ROSs) homeostasis, and (3) cell expansion.

In the history of plant biotechnology, scientists have also carried out intensive histological studies in order to understand the processes taking place inside the callus. One of the first interesting studies carried out using a camera lucida was published in 1923 by Cora Beals [20]. The scientific accuracy of this article remains outstanding today in terms of the interpretation of its results, with the author explaining her hypothesis that shoots must originate from the division of cambium cells. After almost a century, this hypothesis was corroborated by new evidence in molecular biology published by Sugimoto et al. in 2011 [14]. The original study by Cora Beals was later cited by Tsvi Sachs in 1981 [21] to scientifically ground the development process of transport vessels and, furthermore, to hypothesize the need for polar contact with leaves, roots, and a source of auxin for vascular differentiation. This hypothesis is still valid today, after more than 50 years, based on scientific evidence published by Jinwu Deng and collaborators [19].

However, a clear relationship between the callus areas where xylematic elements are developed and de novo shoot development is not yet well established for histological studies. This finding led us to assume that the constant accumulation of scientific evidence could create new opportunities to re-evaluate various old experimental data, to reassess previously obtained scientific results, and to further support new theories [12,14,19]. Therefore, such studies presenting histological analyses of plant callus could now be revised to re-evaluate their scientific results and, furthermore, to provide new insights with respect to the current scientific knowledge. For example, de novo shoot formation from callus should also be assessed as a process based on the pluripotency that is expressed in the callus cells and that is highly dependent on microenvironmental factors (i.e., genetic factors as well as chemical and physical factors from outside the inoculum) [14].

This should be relevant if such studies are to be targeted at the species level. Therefore, we consider that a further analysis of callus's histological sections could provide new insights to better understand de novo shoot organogenesis and, furthermore, to support the connectivity between this process and the development of xylematic elements.

A species of high interest for plant biotechnology at the global level is *Syngonium podophyllum* Schott., which was first micropropagated by Lynn Miller and Toshio Murashige in 1976 [22]. The term for the genus *Syngonium* was first used in 1829 by H.W. Schott; later, in 1851, the same author described *S. podophyllum* as a species of the Araceae family originating in Central America. The species was described as a liana with sagittate- or hastate-shaped leaves located in tropical rainforests. In 1981, Thomas Croat described the geographical distribution of this species as ranging between Mexico, Brazil, Guiana, and Bolivia [23]. For more than 50 years, the species has also spread in Africa, Australia, and Asia, according to the distribution map published by the Global Biodiversity Information Facility [24].

The increase in commercial demand worldwide creates the opportunity for this species to be studied and introduced for industrial micropropagation [25] and, furthermore, to be considered one of the most valuable indoor ornamental plants in the world [26]. Moreover, today, it is known that syngonium is also a source of valuable phenolic compounds with different potential uses in medicine [27–29]. Generally, a major consequence of intense trade is to facilitate the spread of species to new environments. Therefore, today, this species is considered to have a moderate-to-high potential risk of invading new territories and threatening the conservation of biodiversity in newly occupied ecosystems [30].

In Romania, our laboratory has been interested in developing an industrial protocol for *Syngonium* micropropagation since 1996. However, as a former communist country, the lack of access to scientific literature created the opportunity to study this species as completely new for biotechnology. The first goal of this article was to present a series of principles to follow in order to develop an industrial-scale micropropagation protocol. The second goal was to review the whole technological flow for micropropagation stage by stage, based on the previous experience of our laboratory, with the main focus on the formation of callus and de novo shoots. Thus, we reviewed the main results of experiments conducted in our laboratory on *Syngonium* between 1996 and 2004 and published after 1998,

in accordance with the current scientific knowledge, to provide new insights specifically related to de novo shoot formation from callus. Our third goal was to present the results of a histological study involving morphogenetic calli. Particular emphasis was placed on the study of callus areas occupied by xylematic elements and organogenesis (i.e., the formation of shoots, roots, and protocorms).

## 2. Materials and Methods

This article is a review of the main scientific articles published regarding the study of the different stages required for in vitro micropropagation of *Syngonium podophyllum* Schott. cv. 'White Butterfly' by our laboratory, and with reference to the current literature, in our attempt to improve our understanding of de novo shoot formation from callus.

This article is structured in three parts. The first part is dedicated to discussing the main principles followed for setting an industrial micropropagation protocol, along with underlying opportunities and obstacles based on our experience.

In the second part, we further discuss the materials and methods applied for developing an industrial micropropagation protocol, as described in three published articles [31–33] as well as in a doctoral thesis [34]. All three scientific articles were published in the English language—two of them in Romanian journals and one in a journal from the United States. The thesis was published in the Romanian language. All stages—including the starting plant material, sterilization, meristem culture, initiation, micropropagation, and acclimation—are discussed with respect to the existing scientific knowledge on the species. Wounding stress, callus formation, and histological studies of calli are discussed as relevant in order to further contribute to our understanding of the process of de novo shoot formation in syngonium.

In our search strategy implemented in Google Scholar, we included all scientific articles studying the species *Syngonium podophyllum* whose full text was freely available in the English language [35]. In this strategy, the year 1976 was set as the starting point, when the first published article was recorded [22]. There are also some valuable closed-access articles available online, which are mentioned in this paper as citations in the required contexts. However, in choosing the most relevant scientific articles for discussion in this article, the focus was on the following keywords: callus development, callus histology, and de novo shoot formation.

The scientific names of the plant species were all validated against recognized plant taxonomic databases [36–38].

## 3. Results and Discussion

### 3.1. Principles for Starting In Vitro Micropropagation of Syngonium podophyllum Schott. Cv. 'White Butterfly'

*Syngonium podophyllum* Schott. cv. 'White butterfly' was studied as a representative of the Araceae family in Romania for 8 years, between 1996 and 2004. The objective was to develop a micropropagation protocol for industrial purposes, initiated in 1996 as an important goal to increase the indoor plant trade offerings of the Glasshouse Complex Codlea in Brasov County, Romania. This complex functioned for 20 years between 1988 and 2008, after which it was forced to close due to a nationwide economic crisis. It should be noted that the profit of the state-owned company was outstanding, and the decline was mainly the result of political issues [39,40].

In implementing an industrial micropropagation protocol for *Syngonium*, we followed three major principles: (1) to maintain long-term genetic stability for all micropropagation stages, (2) to provide the most cost-effective technology, and (3) to constantly upgrade the technology according to the latest scientific achievements.

To adhere to the first principle, based on our laboratory experience, we used (a) meristem culture to avoid contamination by viruses and other microorganisms (this protocol had already been implemented for carnations, among other species), (b) the simplest and cheapest possible culture media and avoidance of long-term propagation (i.e., by using

low hormone quantities and avoiding mutagenesis), and (c) a very rigorous control of technological factors—such as light intensity and photoperiod, day/night temperature, humidity, air sterilization, and ventilation—similar to existing technology for carnation micropropagation, and for the entire workflow from laboratory to greenhouse.

The second principle aimed to obtain the simplest and most cost-effective industrial protocol possible based on the available reagents in the laboratory and greenhouse, as well as the skills developed by laboratory personnel. A highly rigorous control system was implemented for all technological factors (e.g., light intensity and photoperiod, day/night temperature, humidity, air sterilization, ventilation, electricity supply, heating/cooling during the winter/summer seasons, and disease/pest control in the greenhouse). We also applied rigorous control of all personnel activities and created bypass-type plans for potential technological remedies (e.g., virus detection in case of infection in the laboratory, disease/pest control in the greenhouse, contingencies in case of delays to the import of reagents or other materials).

The third principle, referring to the constant upgrading of technology according to the latest scientific achievements, was implemented with the limitation of restricted access to the newest scientific information due to prohibitive costs. We should also mention the total lack of access to the scientific journals published between 1996 and 2000, during which time the scientific literature library of the Romanian Academy was not open-access. In this regard, three relevant peer-reviewed scientific articles were published in the English language on subjects related to the callogenesis and micropropagation of the species [31–33], which scientifically substantiated a Ph.D. thesis that was publicly defended after six years of research in 2002 [34].

These principles discussed above are consistent with those applied and discussed by other authors [41,42].

In order to develop a completely new micropropagation protocol for newly introduced ornamental species, it was considered relevant at the time to devise some experimental tests for ensuring the most appropriate balance between auxin and cytokinin, as well as the best mineral and vitamin compositions for use in the culture media. In this case, it was preferred to start using the plant growth regulators already available in our laboratory, such as benzyl-aminopurine (BAP), indole-3-acetic acid (IAA), naphthyl acetic-acid (NAA), and 2,4-dichlorophenoxyacetic acid (2,4-D). All of the other reagents (i.e., minerals, vitamins, sucrose, and agar) were provided by Merck, Sigma, and Difco, and they were already in use in our laboratory in the industrial protocol for carnation micropropagation (*Dianthus* sp.), as well as other species, cultivars, and hybrids (i.e., *Gerbera x hybrida* Hort., *Chrysanthemum × morifolium* (Ramat.) Hemsl., *Nephrolepis exaltata var. hirsutula* (*G. Forst.*) *Baker*, sp., *Cymbidium floribundum* Lindl., *Saintpaulia ionantha* H.Wendl., *Sequoia gigantea* Endl., *Solanum tuberosum* L.). Based on the experience of our laboratory, these reagents were considered to be the most reliable in terms of costs as well as for midterm preservation. The simplified micropropagation protocol was published in *Aroideana*—the Journal of the International Aroid Society—in 2004 [33].

*3.2. Micropropagation Protocol Review*

3.2.1. In Vitro Culture Initiation

Four-year-old, healthy, certified mother plants of *Syngonium* imported from the Netherlands were used as donor plants. These plants were maintained in culture pots under controlled greenhouse conditions for more than 2 years to ensure that there was no phytosanitary contamination. It should be noted that using healthy mother plants is essential for starting plant meristem cultures according to Toshio Murashige and Folke Skoog [43]. Explants as stem fragments comprising the nodal segment (i.e., less than 5–6 cm in length) were well washed under running tap water for 2 h. Nodal explants were further immersed in an aqueous solution of Tween 20 (2–3 drops/L) under continuous shaking. After 10 min, all nodal explants were surface-sterilized in the laminar hood flow for 30 s using a sterile solution of 0.1% $HgCl_2$, followed by three consecutive rinses in sterile water for 10 min,

each under continuous shaking. Each nodal explant was placed on sterilized filter paper before taking the meristem under stereomicroscope [33,34].

Sterilized meristems were cultivated on a solidified, modified Murashige and Skoog (MS62) [43] culture medium and sub-cultivated after 10 days under the same conditions to prevent browning of meristems, implementing good laboratory practices to ensure a virus-free process similar to that used for carnations [44]. At this stage, the MS62 inoculation medium was supplemented with BAP (30 mg/L), IAA (1 mg/L), and sucrose (30 mg/L). Merck agar (7 g/L) was added before adjustment of the pH to 5.8, and after sterilization it was reduced to 5.4. The success rate for initiating meristem cultures was 100%, with all of the inoculated meristems being viable [33,34]. This was also due to the disinfecting method used, as well as to the skills of the staff members, who had almost 16 years of practice performing this activity. We should also note that some authors have successfully used antibiotics for in vitro micropropagation of *Syngonium* into liquid culture media at an industrial scale [45].

Meristematic domes with the first or second leaf primordia at a maximum size of 0.2 mm were taken under stereomicroscope in sterile conditions during springtime. *Syngonium* in vitro cultures have also been used by other professional groups [22,46–63] (see Table 1). In 1976, Lynn Miller and Toshio Murashige published the first protocol for micropropagation of *Syngonium*, starting with 0.2–0.4 mm axillary meristems taken under stereomicroscope [22].

**Table 1.** Balance of plant growth regulators and relevant observations related to in vitro micropropagation of *S. podophyllum* from our review of scientific articles whose full text was freely available on Google Scholar.

| References | Plant Growth Regulators for Initiation (mg/L) | Plant Growth Regulators for Multiplication (mg/L) | Observations |
| --- | --- | --- | --- |
| Miller and Murashige 1976 [22] | 3 mg/L IAA + 2 mg/L 2iP | 2 mg/L IAA + 30 mg/L 2iP | No callus described; solidified and liquid culture media; complete technology described |
| Scaramuzzi et al., 1992 [46] | 1 mg/L IAA + 5 mg/L Kin | 1 mg/L IAA + 5 mg/L BAP | Callus formation induced by high levels of cytokinins; cytological and histological studies; complete technology described |
| Salame and Zieslin 1994 [47] | 3 mg/L IAA + 2 mg/L 2iP | 2 mg/L IAA + 30 mg/L 2iP | Peroxidase analysis for in vitro plant wound-stress study |
| Watad et al., 1997 [48] | 3 mg/L IAA + 2 mg/L 2iP | 2 mg/L Kin | Using interfacial membrane rafts for liquid culture media |
| Rajeevan et al., 2002 [49] | 0.5–2 mg/L BAP | 2 mg/L BAP + 0.5–2 mg/L Kin | Callus was also induced at high cytokinin levels |
| Chan et al., 2003 [50] | Not specified | 2 mg/L BAP or 2 mg/L IBA + 2 mg/L BAP | No callus described |
| Schwertner and Zaffari 2003 [51] | 1 mg/L BAP + 1 mg/L IAA | 1–4 mg/L BAP | The best multiplication rate for 4 mg/L BAP; callus was obtained when 4 mg of BAP was added |
| Hassanein 2004 [52] | 1 mg/L BAP | 1 mg/L BAP | Shoot multiplication |
| Chen and Henny 2006 [53] | Citing Miller and Murashige 1976 [22] | Citing Miller and Murashige 1976 [22] | Review of scientific literature on the micropropagation of *Syngonium* |
| Zhang et al., 2006 [54] | 80 mg/L adenine | 0.2 mg/L NAA + 2 mg/L BAP | No callus formation; the study is relevant for somatic embryogenesis |

**Table 1.** *Cont.*

| References | Plant Growth Regulators for Initiation (mg/L) | Plant Growth Regulators for Multiplication (mg/L) | Observations |
|---|---|---|---|
| Wang et al., 2007 [55] | 80 mg/L adenine | 0.2 mg/L NAA + 2 mg/L BAP | No callus formation; the study is relevant for somatic embryogenesis |
| Cui et al., 2008 [56] | 0.1 mg/L NAA + 0.2 mg/L TDZ | 1 mg/L NAA + 2 mg/L CPPU or 2 mg/L TDZ | Callus description; protocorms and histological study; the study is relevant for industry |
| Rajesh et al., 2011 [57] | Not specified | 20 mg/L BAP | The average shoot formation was similar to our results: 9.5 shoots/explant |
| Teixeira Da Silva et al., 2014 [58] | Not specified | Not specified | Ploidy study of in vitro plantlets |
| Kalimuthu and Prabakaran 2014 [59] | 0.5–3 mg/L BAP + 200 mg/L NaH$_2$PO$_4$/0.2 mg/L NAA/0.2 mg/L TDZ | 0.5–3 mg/L BAP + 200 mg/L NaH$_2$PO$_4$/0.2 mg/L NAA/0.2 mg/L TDZ | The best results were obtained for the combination 1 mg/L BAP + 200 mg/L NaH$_2$PO$_4$ |
| Teixeira Da Silva 2015 [60] | Citing Wang et al., 2007 [55] | Citing Cui et al., 2008 [56] | Ploidy study of in vitro plantlets |
| Moumita et al., 2016 [61] | MS62 | 2 mg/L BAP + 0.5 mg/L NAA | The best formula for shoot multiplication |
| Kane 2018 [62] | Citing Miller and Murashige 1976 [22] | Citing Miller and Murashige 1976 [22] | The chapter provides an activity for students' education |
| Sharifi et al., 2022 [63] | MS62; not in English | 1 mg/L BAP + 3 mg/L Kin | The authors focused on testing shooting success |

Abbreviations: IAA: indole-3-acetic acid; 2iP: 6-($\gamma,\gamma$-dimethylallylamino)purine; Kin: kinetin; BAP: 6-benzylaminopurine; IBA: indole-3-butyric acid; NAA: 1-naphthaleneacetic acid; CPPU: (2-chloro-4-pyridyl)-N′-phenylurea; TDZ: thidiazuron, or N-phenyl-N′-1,2,3-thiadiazol-5-ylurea.

Further stages in the development of meristems are discussed in our previous papers [33,34]. After 12 weeks of cultivation in the same culture medium, it was generally possible to consider that all inocula were sufficiently developed (i.e., a diameter ranging between 0.5 and 1 cm) to the first multiplication stage by splitting clusters of buds into two to four pieces. The first multiplication stage was carried out in MS62 culture medium supplemented with BAP (1 mg/L), Kin (3 mg/L), and IAA (0.1 mg/L). After a total of 20 weeks from the initiation of the process (i.e., the day of meristem inoculation), it was possible to isolate clearly differentiated shoots (i.e., heights ranging between 10 and 15 mm) to begin the multiplication experiments, and only these shoots were recorded to assess the multiplication rate. All shoots and buds less than 10 mm in height were considered to be insufficiently developed to enter the economic workflow at the industrial scale. Consequently, these calli were transferred to new culture media in order to ensure further development of buds and elongation of shoots.

The balance of plant growth regulators used in our laboratory for meristem cultures was in favor of cytokinin (i.e., 30 mg of BAP and 1 mg of IAA) [33,34]. Other authors used different balances of plant growth regulators, as well as different regulators. For example, Lynn Miller and Toshio Murashige used a hormone balance in favor of auxin (i.e., 2 mg of 2iP and 3 mg of IAA) [22]. Other teams used a balance in favor of cytokinin [49,51,52,56–59]. Later, other authors used new synthetic regulatory substances, again more inclined towards cytokinin, such as 1 mg of NAA and 2 mg of (2-chloro-4-pyridyl)-N′-phenylurea (CPPU) or 2 mg of N-phenyl-N′-1,2,3-thiadiazol-5-ylurea (TDZ) [56,59] (see Table 1).

As shown Table 1, very different types and balances of plant growth regulators have been used in different laboratories to initiate meristem cultures, indicating that this species

is very reactive towards different culture conditions and very easy to culture in vitro. We should also note the contributions to developing a micropropagation protocol by relevant researchers who are cited by others but whose articles were not available. For example, Scaramuzzi and coworkers [46] cited an unavailable paper published by Makino and Makino in 1978 (e.g., the article was entitled "Propagation of Syngonium podophyllum cultivars through tissue culture", and published in In Vitro, volume 14, page 357). Some other articles were also unavailable or written in languages that do not use the Latin alphabet.

The decision to incline the balance of the plant growth regulators in favor of cytokinin was based on our observations related to the well-expressed apical dominance of the main shoots of the pots' donor plants. Apical dominance has long been known to be supported by the high internal levels of auxin secreted by young organs [64]. However, the interruption of this auxin gradient flow can be achieved during the cutting-off of the meristem, and this process is confirmed by the further addition of auxin to the meristem's culture medium, as reported by Lynn Miller and Toshio Murashige [22]. This idea was not tested in our laboratory, and it should be taken into consideration for further studies.

3.2.2. Micropropagation and Culture Media Testing

In the 28th week of in vitro cultivation, we began the testing of auxins and vitamins using an MS62 basal medium composition [33,34]. The highest recorded multiplication rate was ~9.17 shoots/explant in the case of an auxin–cytokinin ratio of 0.5 mg/L NAA to 1 mg/L BAP. It should be noted that only shoots over 7–9 mm in height were recorded.

In the 36th week of in vitro cultivation, we conducted the cytokinin tests, and based on our results the rate of multiplication was slightly higher than 9.85 shoots/explant [33,34]. This multiplication rate was slightly higher than that obtained on the same solidified MS62 culture medium by Lynn Miller and Toshio Murashige (i.e., 7.9 shoots/explant, with their minimum height not defined) [22]. However, Miller and Murashige obtained a higher multiplication rate when liquid culture medium was used (i.e., 26 –> 60 shoots/explant). Comparable multiplication rates were also reported by other authors [46,49,51,53,57,59,61,63].

In our laboratory, a multiplication rate of 9 shoots/explant was considered to be excellent for solid culture medium use when recording only shoots of over 7–8 mm in height, and we also found a low capacity for infection compared to liquid culture media. At the end of this stage, the height of the shoots used for industrial-scale micropropagation was set to be longer than 7–8 mm. Other authors obtained the same multiplication rate using different compositions of plant growth regulators and different cytokinin–auxin ratios [57]. The clusters comprising shoots shorter than 7 mm in length, as well as buds, were transferred for further multiplication and elongation of the shoots on the same culture medium, with a decreased cytokinin–auxin ratio [33]. However, increasing the shoots' multiplication rate to more than 60 shoots/explant should be more cost-effective for starting the technological flow and for producing a large number of shoots in a very short time. Some other teams were also successful in obtaining somatic embryos, completing all morphogenetic programs for the species, but without callus generation [54,55].

A multiplication rate of nine plantlets/explant up to the greenhouse stage generated total costs of 1 USD/100 pot plants, and in 1998 this was considered to be profitable. Under present conditions, such a value is no longer worthwhile; furthermore, the technology needs to be updated for the new energy requirements. Starting with the third stage of the technological flow, different experimental tests were initiated. Relevant stages related to Syngonium micropropagation are presented in Figure 1.

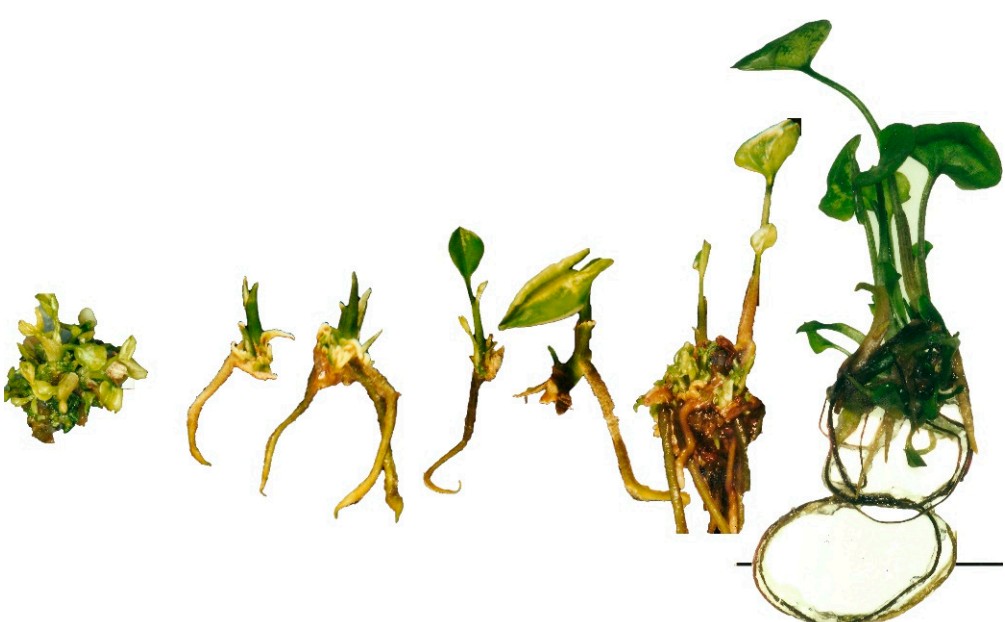

**Figure 1.** Different stages in the micropropagation of *Syngonium podophyllum* cv. 'White Butterfly'. From left to right, de novo shoots over 7 mm in height were separately transferred to solidified MS62 culture medium for elongation of shoots up to the moment of transfer for acclimation. The culture medium's composition is described in 2004 [33].

### 3.3. Callus Induction and Development

### 3.3.1. Wound Stress and Callus Initiation

In 2000, based on contract no. 6127/2000 signed with the Romanian Ministry of Research, it was possible to develop an experimental test to understand the relationship between wound stress and callus initiation for shoot formation. The expected results were to be used to improve the *Syngonium* micropropagation protocol [33,34]. In that experiment, we used fragments of in vitro petioles and leaves, and salicylic acid (SA) and the synthetic hormone 2,4-D were tested as signaling molecules. SA was recognized at the time for its role in preventing wound signaling in plants [65,66], while 2,4-D had long been recognized for its role in inducing callus formation based on the experiment conducted by John Torrey and Kenneth Thimann in 1949, who sprayed this herbicide on the surface of stumps of the tropical sicklebush tree (*Dichrostachys nutans* Benth.) [67]. They observed that a ring of callus tissue developed at the cambium level after a couple of weeks. We should also note that the use of the synthetic plant growth regulator 2,4-D for in vitro plant experiments was successfully applied after 1945 [68–70].

Today, it is already recognized and scientifically proven that wound stress is responsible for the production of callus tissue in order to seal the wounded site and to facilitate adventitious organogenesis or somatic embryogenesis [19].

### 3.3.2. Experimental Design

The tested culture media were composed of two parts: the first part was a solidified MS62 culture medium supplemented with BAP (1 mg/L), which was covered with the second part, comprising 2 mL of liquid culture medium with the same composition and supplemented with 2,4-D (0.1 mg/L) and/or SA (0.1 mg/L). The entire experiment was set up in Petri dishes of 6 cm in diameter. The inoculum was fixed to the surface of the solidified culture medium, and a thin liquid culture medium covering the inoculum ensured rapid and uniform contact with the used signaling molecules.

### 3.3.3. Total Peroxidase Activity

The total peroxidase activity (TPA) was assessed at 72 h according to the method of Dana Iordăchescu and Ioan Florea Dumitru published in 1988 [71]. The size of the leaf fragments was 0.5 cm × 0.5 cm, while the petiole fragments were 1 cm in length. As a general control, unwounded shoots were used. In this case, our experimental hypothesis was based on the knowledge that wound stress is already expressed during in vitro cultivation of plants, according to various authors [72,73]. Moreover, it has long been observed that a burst of phenolic compounds is generally released at the wound sites of plants, and that this process is immediately followed by the activation of membrane-bound peroxidases that are required to initiate healing processes [65,74].

- Effects of 2,4-D:

After 72 h, the petiole fragments treated with 2,4-D exhibited a 1.84-fold increase in TPA compared to control shoots (i.e., intact shoots). In this case, the auxin and the supplemented cytokinin clearly contributed to an increase in TPA to initiate cell proliferation, as the first and most obvious event was recorded four weeks later. In the case of control petioles, we found a 1.6-fold decrease in the TPA activity compared to explants in the presence of auxin, suggesting that auxin activated peroxidase activity in the petiole fragments. In fact, upon the increase in TPA activity at the wound site, cell proliferation began as an essential process for the survival of the plant inoculum. The calli or fragments of plants can further act to continue the plants' morphogenesis [10]. In the case of leaf fragments, 2,4-D induced a 2.45-fold increase in TPA activity compared to intact leaves and a 1.25-fold increase compared to the control leaf fragments. Due to the larger leaf surface for wounding, the release of phenolic compounds [27–29] might have been responsible for the activation of enzymes with peroxidase activity, which should greatly enhance the wound stress response of leaf fragments in the presence of 2,4-D.

- Effects of salicylic acid:

For petiole fragments, SA induced a 4.26-fold increase in TPA activity compared to petioles taken from intact shoots and a 1.46-fold increase compared to the control petiole fragments. This value was more than double the effect of 2,4-D at the wound site and is consistent with previous scientific results [60]. In the case of leaf explants, SA induced a ~1.2-fold increase in TPA activity compared to the intact leaves but, conversely, a 1.6-fold decrease compared to control leaf fragments. Here, SA induced TPA activity at half the rate of 2,4-D. Moreover, the TPA value in fragmented leaves was almost equal to that in intact leaves from intact shoots. It can be concluded that the exogenously applied SA may minimize the expression of TPA in syngonium at the leaf level, but it can induce overexpression in the fragments of petioles.

- Effects of 2,4-D and salicylic acid:

When both compounds (2,4-D and SA) were exogenously applied, it appeared that the TPA activity was similar to that obtained in the fragmented petioles or leaves subjected only to 2,4-D. Thus, the effects of SA appeared to be hidden by the presence of 2,4-D.

### 3.3.4. Callus Initiation from Fragmented Petioles and Leaves

The entire experiment was observed continuously for 8 weeks. In the first three weeks, leaf and petiole fragments subjected to 2,4-D became etiolated compared to controls and those treated with SA, which remained green. All petiole fragments subjected to SA developed hypertrophy at the cut part of the petiole following the circular cambium, which also spread along the entire length of the fragment. However, some of these samples only developed callus at the wound site—white and friable, in small quantities—and later in the cambial tissue as well. Among the two types of explants, only petiole fragments were able to produce a proliferative callus for syngonium at the basipetal part of the petiole, in a similar manner to that described for other species [62]. In contrast, based on the conditions of this experiment, it was not possible to induce callus formation at the leaf fragment level.

Moreover, other authors were successful in initiating and studying callus formation based on a specific hormone balance [46,49,51–53,56,63]. However, it should be noted that a burst of phenolic compounds may also affect callus formation [75].

In another experiment conducted by Jin Cui and collaborators on the same cultivar but under different culture conditions (i.e., different plant growth regulators and different balances), the proliferation of callus was also possible at the leaf level [56].

Re-evaluating these results, it can be concluded that wound stress is also responsible for a clear sequence of events, i.e., increased peroxidase activity, followed by the expression of hypertrophy at the wound sites and, subsequently, by callus development. Hormone balance and mineral composition are the major inductors of callus formation, and different organ fragments may respond differently. The lack of callus formation in fragmented leaves under our experimental conditions may have been due to the culture medium composition, which was not supportive of the expression of hypertrophy, although it was proven to express the highest peroxidase activity. This subject should be of interest when investigating callus initiation for different purposes [46,49,51–53,56,63].

*3.4. Review of Histological Studies on Syngonium Calli Obtained with Different Media Compositions*

In 1998, the very first scientific article dedicated to the study of the effects of medium composition on in vitro morphogenesis was published in Romania in the English language [31].

The first hypothesis of the experiment was based on the statement that morphogenesis in plants is a very complex process that is strictly controlled in time and space by its genetic and external factors [76]. In this regard, we considered that a histological study was needed to reveal morphogenetic events such as de novo morphogenesis of shoots and roots—as proposed for other species by various authors—to better describe and understand the in vitro morphogenesis of syngonium [77,78]. To this end, young shoots of 8–10 mm in height were tested by cultivating them in two types of cultivation media—MS62 [39] and Nitsch 1969 (N69) [79]—as basal mineral compositions [31,34].

The second hypothesis considered that, with *Syngonium* being a tropical plant, it would be interesting to study the effects of a completely different mineral composition in parallel to that of the MS62 culture medium. We considered the fact that essential elements would be at different concentrations in the original tropical rainforest environment. Therefore, for successful in vitro multiplication at a low cost, it was also considered relevant to test various culture media whose mineral composition could be modified. At the time, we also considered that the electrolytic strength of the MS62 culture medium should be different compared to that of N69. Furthermore, it may play an important role in organogenetic processes such as de novo shoot formation and rooting (i.e., different osmotic pressure). In the case of N69′s mineral composition, calcium was at half the usual concentration, while zinc and manganese were increased (from 8.6 to 10 mg/L and from 16.9 to 18.94 mg/L, respectively). In addition to these changes, cobalt and iodine were absent, boron was supplemented to an increased concentration (from 6.2 to 10 mg/L), and the concentrations of phosphate and nitrate were reduced (from 85 to 68 mg/L and from 825 to 720 mg/L, respectively). Later publications provided highly accurate descriptions of relevant differences in mineral changes for tropical forests [80,81].

3.4.1. Experimental Design and Histological Method

The basal mineral culture media (i.e., MS62 and N69) were supplemented only with MS62 vitamins and the same hormonal balance: 2,4D (0.1 mg/L), BAP (1 mg/L), and Kin (3 mg/L). The molar ratio was inclined towards cytokinin which, according to our previous experience, was needed to test the morphogenetic process. Based on the aforementioned working hypotheses for implementing this experiment, the morphogenetic callus obtained after 8 weeks of cultivation was processed for histological study.

For histological analysis, all callus samples were collected at the same time and immersed in Navashin's fixation solution at room temperature, as recommended by other authors [82], and then in liquid paraffin. After the samples solidified, they were sectioned at 8–10 μm and colored on slides with a solution of hematoxylin–eosin in order to enhance the contrast between different tissue structures [83].

The published conclusions of this experiment revealed that different medium compositions induced distinct responses in syngonium callus organogenesis: MS62 culture media induced the formation of a softer, yellowish callus expressing complete organogenesis (i.e., shoots and roots), as opposed to N69, which induced the formation of a harder, greener callus expressing only shoot formation [31]. In 1992, Scaramuzzi and collaborators in 1992 published the first histological analysis of the morphogenetic callus, the findings of which were similar to our results [46]. Later, a similar yellowish callus was also obtained and described by Jin Cui and collaborators [56].

3.4.2. Histological Analysis of Callus: Xylematic Elements and Protocorms

The histological analysis of calli obtained in MS62 and N69 culture media revealed some interesting peculiarities that should be further discussed. In 1998, we underlined the significant differences that appeared for organogenetic responses in relation to "xylematic-like structures" (XLSs) detected by our team in the structures of the calli [31]. As we now have full access to additional scientific articles, we felt it important to investigate the scientific meaning of the "xylematic-like structures" once more. We found that, in 1995, such structures had already been described as procambial cells for *A. thaliana* by Simona Baima et al. [84]. The authors proved at the time that the auxin IAA involved in vascular development modulates the expression of *Athb-8*—a gene responsible for the regulation of vascular development in *A. thaliana*. In that period, the term "vascular bundle" was well established, having also been coined early in 1920 for histological studies of callus by Robert John Harvey-Gibson and Elsie Horsman in Liverpool (UK) [85].

The scientific quest to understand how vascular bundles or strands are formed yielded more results after 1980, and yet some 20 years ago the mechanism was still largely unknown [86]. In 1987, Harry Klee and collaborators stated that auxins are able to induce differentiation of xylem tracheary elements (XTEs) in suspension cultures of certain species [87]. This is also consistent with our observations of syngonium, as we also used the term xylematic-like elements (XLEs) for tracheary elements observed on histological slides, as a precautionary step. However, the fact that auxins trigger vascular bundle formation was scientifically established in 2000 [88]. The potential role of XLEs had not been discussed previously, but in trying to explain the hard consistency of the callus obtained in N69 culture medium (mineral composition), we consider that this revision is needed.

A closer analysis of the slides for the softer calli obtained in the MS62 culture medium (see Figure 2) clearly reveals that they are of smaller dimensions and constantly appear in the vicinity of organogenetic polarized structures and embedded in the larger parenchymatic cells. The XLEs may be considered to be protoxylem elements—a topic that had not been discussed previously—but it is obvious that they appear constantly in the vicinity of de novo shoots and roots, as well as that of the nodular-like structures. This constant positioning of the XLEs may be due to their potential role in supporting the development of new organogenetic structures. Their high density around the polarized organogenetic structures may support this hypothesis, as they are not observed in callus zones where organogenesis is not occurring. Moreover, this is consistent with the study by Thomas Berleth and collaborators published in 2000 [88], as well as considering the role that auxin may play in the establishment of plant cells' polarity and oriented differentiation which, in turn, are needed for aligning vascular differentiation, among other functions. Following this logic, XLEs develop in the parenchymatic tissue of the callus to support the further development of organogenetic centers in order to accomplish their final morphogenetic objective: whole-plant formation. This hypothesis is supported by recent results that are based on the analysis of molecular markers, supporting the idea that a callus is a

group of pluripotent cells where—under appropriate conditions—the promotion of auxin self-production and the enhancement of cytokinin sensitivity are also required for organogenesis [89]. In our case, culture conditions ensured the availability of auxins as well as that of cytokinin. The left-hand image in Figure 2 presents a cross-section of the upper part of the shoot tips, as well as other organogenetic areas, in circular shapes that may occur at the basal parts of different organogenetic areas of the callus. Similar images were published by Scaramuzzi [46], further substantiating our results.

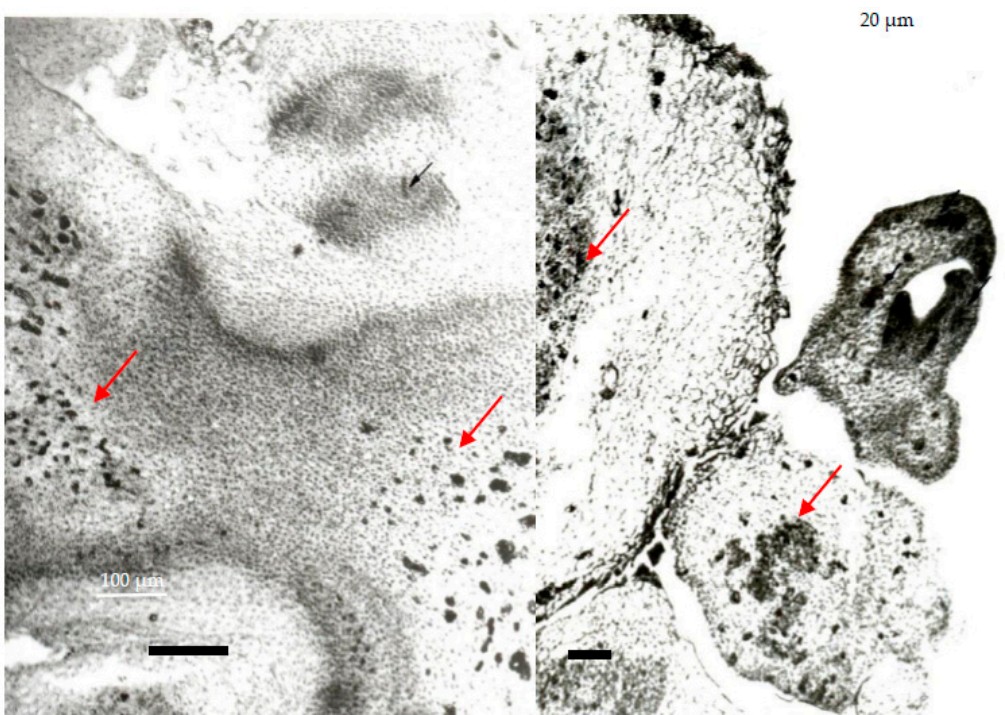

**Figure 2.** Histological analysis of slide images taken for *S. podophyllum* callus obtained in the MS62 culture medium. This was a full organogenetic callus producing roots and shoots. Nodular-like structures where organogenesis is taking place can be observed (in the tips of the right-hand shoot formation and in a transversal section of the upper part of the left-hand shoot formation; black arrows). Xylematic-like elements or bundles can be observed in the vicinity of all organogenetic centers (see red arrows; magnification ×270 (left) and ×135 (right); bars = 100 µm).

By changing the basic mineral formula, e.g., for the N69 mineral composition, the harder greener callus revealed the formation of well-developed XTEs during the histological study (see Figures 3 and 4). By analyzing different histological images taken at the time, we further observed that these XTEs were constantly arranged in areas positioned at the bases of meristematic domes, as well as those of developing leaves or shoots. Small XLSs could also be observed close to areas comprising polarized nodular-like shapes recognized for their relevance in the initiation of organogenesis.

A re-evaluation of old slides revealed the presence of protocorms (see Figure 3) that were very well differentiated in a transversal section, which were not mentioned previously [31]. Different-sized images of such protocorms were frequently seen in the slides. It should be noted that protocorms were also described by other authors in different culture conditions [56,60].

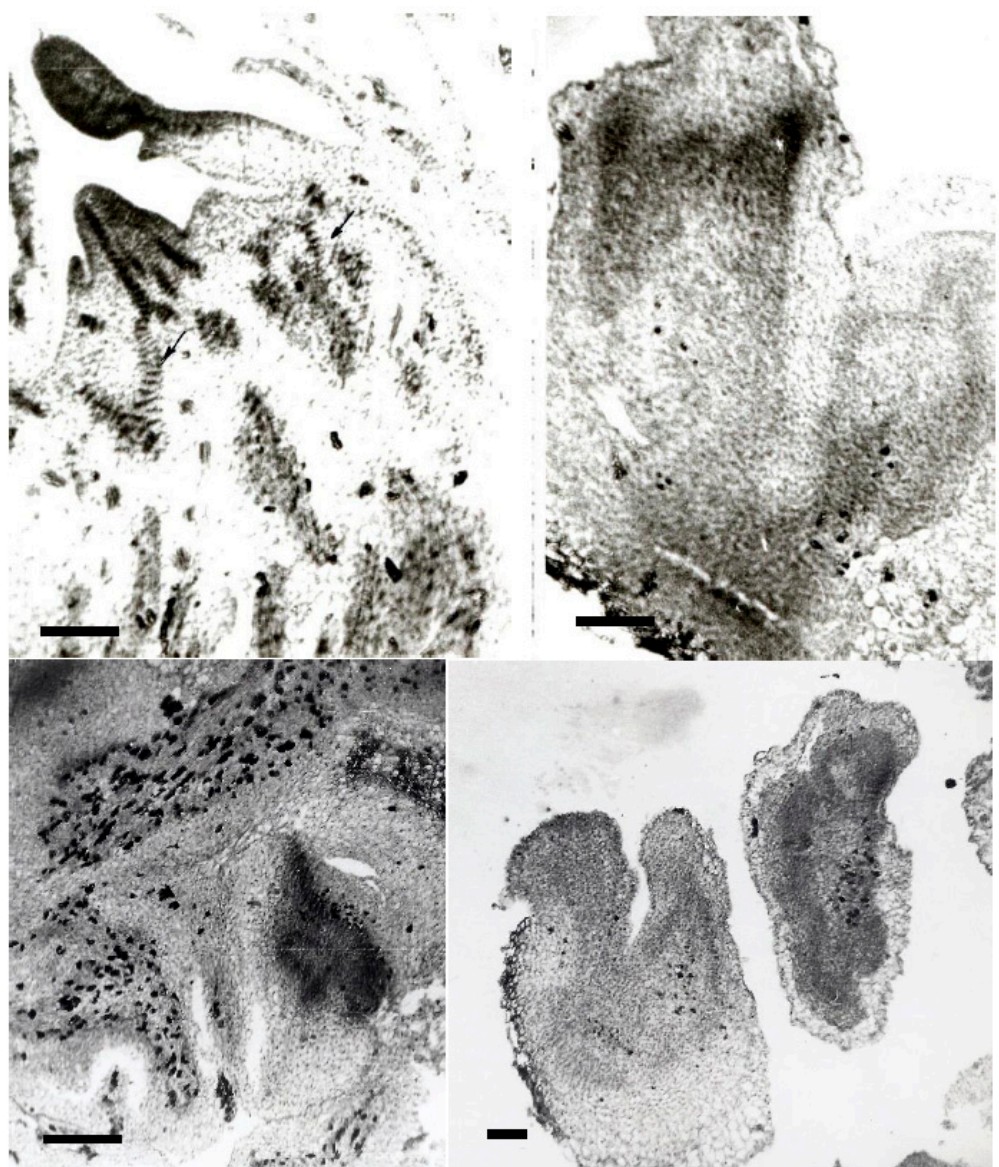

**Figure 3.** Histological analysis of slide images taken for *S. podophyllum* callus obtained in the N69 culture medium. Shoots and protocorms forming callus were observed. Nodular-like structures could also be observed (lower-left image), where organogenesis was taking place. The black arrows indicate very well-developed XTEs in the vicinity of a meristematic dome and de novo shoot formation. The meristematic dome is very well expressed, with well-developed, asymmetric, primordial leaves. In the upper- and lower-right images, protocorms can be observed in transversal section (magnification ×135 (upper and lower left) and ×270 (upper right), bars = 100 μm).

In the left-hand images (Figure 3), a transversal section of a meristematic dome is very clear, and the asymmetry of its contour reveals the initial stages of leaf formation. Again, XTEs are well developed nearby and entering the meristematic dome. Beneath the meristematic area, there are certain callus zones where XTEs reside among parenchymatic cells. No visible root formation was observed for the N69 mineral composition in 1998. Based on the reassessment of these images, it appears that the same auxin–cytokine ratio and the same vitamin composition but different mineral composition can change the pluripotency and totipotency of callus cells. While the MS62 mineral composition supported separate and complete organogenesis (i.e., shoots and roots), rooting was no longer observed when changing to the N69 mineral composition. Instead, protocorms were very well developed and expressed at a high density, appearing in almost half of the investigated images.

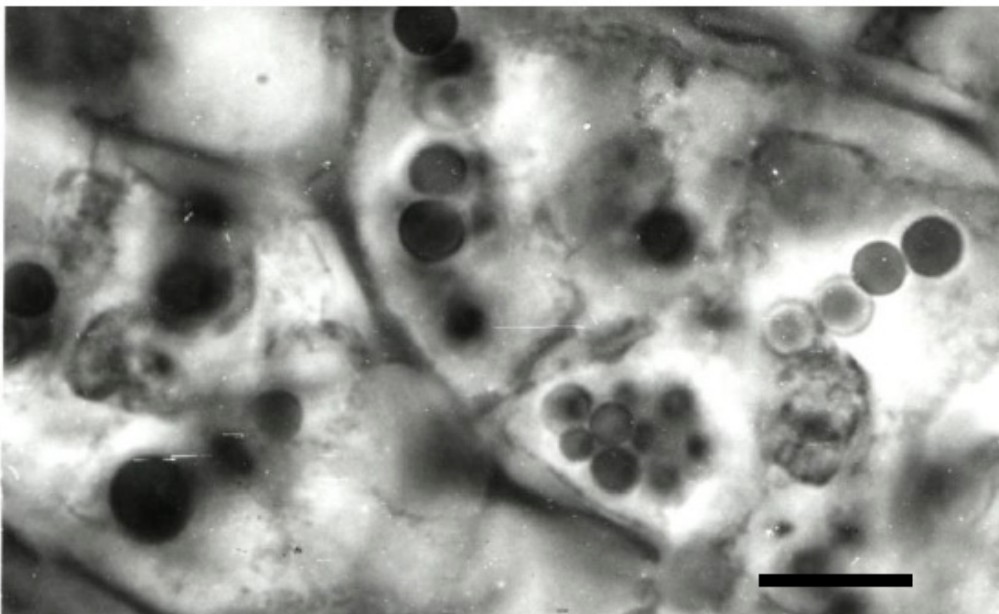

**Figure 4.** Histological analysis of slide images taken for callus cells of *S. podophyllum*. The small cells in the vicinity of the organogenetic areas present many starch inclusions (magnification ×45,000, bar = 10 μm).

Another easily noticed difference when we analyzed the XLEs is that they were less developed in MS62 compared to the N69 mineral composition. Therefore, it might be possible that the lack of or decrease in the concentration of certain mineral elements is responsible for this type of organogenesis, as was already stated some 47 years ago for *Antirrhinum majus* [90]. In another study, the increased concentration of boron appeared to influence shoot regeneration rather than contributing to increasing the number of shoots/explants, taking into account the results of recent studies on date palm [91]. The high expression of XTEs can also be supported by recent findings [92] and, among others, by the recognized toxicity of boron to plants [93]. The Lewis theory regarding boron's regulatory role in lignin synthesis [94]—highly supported by other scientists [95]—seems to be consistent with our findings regarding the cultivation of *Syngonium* callus in N69 culture medium. However, there are also too many other variables related to the changes in the compositions of other minerals, as mentioned previously. The lack of or decrease in the concentration of cobalt in N69 compared to MS62 was recently associated with decreased shooting in *Cucumis sativus* [96]. An interesting experiment was published by Renata Garcia and collaborators in 2011, where they used a mineral composition that was different from the basal MS62 and similar to N69 for callus cultivation of *Passiflora suberosa* [97]. They also observed the formation of a more compact callus and a significant decrease in shoot formation per explant. However, rooting was not impaired for this species.

### 3.4.3. Histological Analysis of Callus: Parenchymatic Cells Full of Starch Inclusions

A closer view of the small cells residing in the parenchymatic callus tissue and located very close to organogenetic centers and XLEs revealed that they contained many starch inclusions (Figure 4). This effect may be related to the high metabolic activity that is required in these zones to support the development of new organogenetic centers or XLEs. This observation further supports the findings of previous authors working on different species [98,99]. Recently, it was proven that there is a close relationship between lignin metabolic processes and the metabolism of starch and sucrose as the main factors associated with callus regeneration [100], which could be further investigated for different processes of cell differentiation.

### 3.4.4. Biochemical Analysis

By analyzing the spectra of electrophoretic peroxidases (POXs) published in 1998, we found clear differences between these two types of callus, further supporting our previous statement that N69 culture medium does not support root organogenesis [31,34]. Based on that analysis, 13 POXs' electrophoretic bands were described for a complete morphogenetic callus obtained in the MS62 culture medium (i.e., roots and shoots). The 6th electrophoretic band belonged to green tissues and the 12th belonged to roots. The 4th, 7th, 11th, and 13th appeared to mark the morphogenetic callus without visible specialized organs. The 3rd electrophoretic band appeared in all samples, and the 12th was the single band that was missing in the callus cultivated in N69 medium [31,34].

The continued development of morphogenetic processes revealed that N69 basal mineral medium was able to produce 42.56 propagules as buds and shoots per explant after 8 weeks of cultivation, while MS62 basal mineral medium induced 26.06 buds and shoots per explant. This result is consistent with the idea that the mineral composition is relevant both for increasing de novo shoot development per explant and for improving shoots' elongation.

### 3.5. Effects of the Vitamins MS62 and N69 on the De Novo Proliferation of Shoots

Another experiment was conducted to further study the effects of the vitamins MS62 and N69 on shoot multiplication and shoot height, as well as the effects of the composition of the plant growth regulators (but using only an MS62 basal mineral composition). Data on this topic were also published in 2004 [33,34]. In this experiment, we used NAA at different concentrations (0, 0.1, 0.5, and 1 mg/L) and a constant concentration of BAP (1 mg/L). The analysis of the results revealed that the best balance of plant growth regulators for the MS62 culture medium was 0.1 mg/L NAA and 1 mg/L BAP, which produced an average of 9 shoots/explant, considering only shoots > 7 mm in height. When increasing the auxin to 1 mg/L, the shoot multiplication rate decreased significantly compared to the control (i.e., no auxin) and the culture medium supplemented with 0.1 mg/L NAA. In this case, apical dominance of the main shoot was also observed (i.e., an average of 25.5 mm). All the culture media induced the formation of both shoots and roots, as well as a healthy appearance of the shoots. Again, the results of this experiment suggest that the best formula for de novo shoot formation was that provided by the MS62 basal culture medium [34].

### 3.6. Effects of Cysteine on Callus Formation in N69 Medium

A follow-up experiment was carried out to test the effects of cysteine. Figure 5 shows that a greener callus developed on the N69 medium in the presence of cysteine [34]. Moreover, visible root development and shoot elongation were observed in the presence of cysteine. The purpose of this experiment was to understand whether shoot regeneration in this species may be influenced by the potential release of phenolic compounds—as established in other species according to several reports [101,102]. It should also be noted that cysteine has long been recognized for its action against the release of phenolic compounds, and it has been recommended to be used for in vitro micropropagation of plant species [103,104]. Moreover, the fact that *Syngonium* produces phenols was first observed by Lynn Miller and Toshio Murashige [22]. These observations were later confirmed by several teams of scientists, proving that this ornamental species also has a high therapeutic value—mainly due to phenolic compounds with high antioxidant activity [27–29].

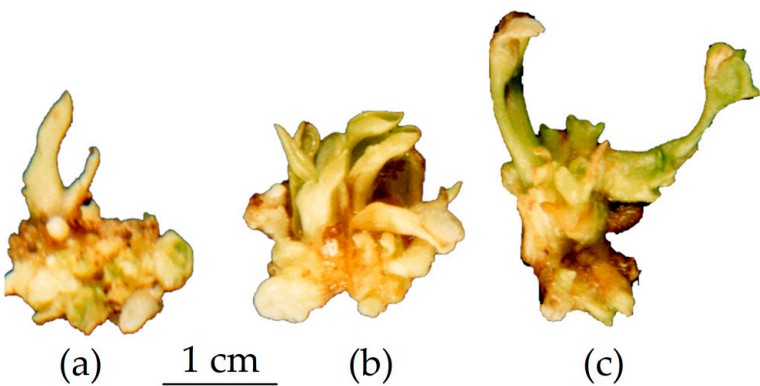

**Figure 5.** Different types of *Syngonium podophyllum* calli: (**a**) Control callus obtained in MS62 medium without cysteine. (**b**) Control callus obtained in N62 culture medium without cysteine. (**c**) Callus obtained in N69 culture medium supplemented with cysteine. It can be seen that the callus was much greener when it was cultivated in N69 culture medium, and that the shoots were more elongated in the presence of cysteine.

Later, in 2001, Jose Carlos Lorenzo et al. proved that the addition of cysteine reduced shoot formation and the excretion of phenols in the in vitro cultivation of sugarcane [105]. These authors also acknowledged several examples that showed contrasting results in developmental processes for other species and in vitro systems. Positive effects for micropropagation were also observed for grapevine (*Vitis vinifera*) [106], air potato (*Dioscorea bulbifera*) [107], and *Prosopis* species [108]. The elongation of shoots was also observed under cysteine treatment in *Prosopis* species [108]. In the case of the common bean (*Phaseolus vulgaris*), shoot elongation was also observed [109].

De novo shoot formation is a complex process that is highly desirable for all in vitro plant systems, especially to reduce costs for industry. *Syngonium* was investigated in vitro by various research teams before and after our publications [22,46–63]. Most of the teams obtained de novo shoots by using the same basal mineral and vitamin composition of MS62, but using different plant growth regulators as well as different balances between cytokinin and auxin.

The histological study of the callus may provide new insights regarding the stages needed for de novo development of the shoots [31,46,48,49,51–53,57,61]. Moreover, by studying the wound stress of the species, we may provide further evidence for the relevance of the hormone balance used for initiating in vitro callus development [32]. The chain of events consisting of increasing the peroxidase activity, hypertrophy of the wound sites, and callus formation starting from the cambium in petioles is already consistent with findings for other studied plant species [19,54].

Whether the callus can be well established in an in vitro culture to produce de novo shoots rests largely on the use of appropriate mineral, vitamin, and plant growth regulator balances. In this regard, the histological analysis of both types of calli that were cultivated—in MS62 mineral composition culture media and in N69 mineral composition culture media—revealed significant differences that were supported by biochemical analysis of peroxidases.

The aforementioned results raise the following question: is it possible for *Syngonium* calli cultivated in N69 media to also assist in the increased accumulation of phenolic compounds responsible for the development of XTEs? Here, we must underline the clear difference between XLEs developed for the calli cultivated in MS62 and those obtained in N69, as observed in all investigated slides. It should be noted that it has already been proven that cysteine plays a direct role in decreasing the accumulation of phenolic compounds and supporting organogenesis in the hybrid *Miscanthus × giganteus* [110].

It has been proven that the addition of an extra nitrogen source can have a positive influence on the development of xylematic bundles and on pigment accumulation in

paperwhite (*Narcissus tazetta*) [111]. The direct relationship between lignin production and the development and maturation of tracheary elements upon apoptosis has also been well documented [112]. Additionally, it is possible that the tracheary elements observed in the tiny calli could be formed as a result of phenol accumulation under more stressful in vitro culture conditions. Moreover, it was observed that an increased thickening of the xylematic vessels of candy leaf (*Stevia rebaudiana*) took place under cold stress [113]. Previous studies also proved that the production of xylematic vessels is stimulated under different stress factors, e.g., physical and chemical [114].

In 2018, Shokoofeh Hajihashemi and Omolbanin Jahantigh put forward the hypothesis that an increase in the development of xylematic vessels may further support water transport, among other processes [111]. They also cited an earlier observation relating to the diameter and frequency of xylem vessels, which are critical determinants of water conductance [115]. In this regard, the clear and consistent positioning of the xylematic tracheary elements close to de novo shoot development areas and other meristematic structures provides rapid access to nutrients and water, as previously stated by other authors [114]. Regarding the positive effects of cysteine in supporting shoot elongation, it should be noted that it was proven to stimulate shoot elongation for *Petunia x hybrida* by acting on the gibberellic acid pathway [116,117]. Conversely, by inhibiting the synthesis of cysteine, the opposite effects were observed in cockspur (*Echinochloa crus-galli*) [118].

All of these findings are relevant to further support the improvement of the technology for industrial-scale *Syngonium* production in a more cost-efficient manner, and theoretical study of de novo shoot formation can contribute to this technological improvement. Cysteine may play an important role in decreasing the effects of phenolic compounds discharged under these stressful conditions when cultivating *Syngonium* in N69 culture medium, as well as improving de novo shoot elongation [119]. Thus, the addition of cysteine in the pre-acclimation phase can more quickly reinforce the formation of more adaptable plantlets for acclimation [120,121].

## 4. Conclusions

*Syngonium* is a very reactive species that is easy to cultivate in vitro. Different balances of plant growth regulators can be used to successfully initiate meristem cultures and the first stages of micropropagation of this species. Moreover, the different vitamin and mineral compositions tested could not completely impede de novo shoot development. All manner of morphogenetic programs have been studied since 1976, including shoot formation, rooting, callogenesis, protocorm formation, and embryogenesis. The main basal mineral medium composition used was MS62. Based on our histological analysis of calli originating from MS62 and N69, some obvious differences were observed in their structure. Thus, we observed the constant positioning of xylematic elements in the callus zones adjacent to meristematic and nodule-like areas, as well as organogenetic centers. Additionally, protocorms of *Syngonium* were identified with N69, and their formation was associated with very well-developed tracheary elements, constituting a novelty of this study. Conversely, the organogenetic centers observed in the callus originating from MS62 revealed less-differentiated xylematic elements, but these were also constantly in the presence of and closely connected with meristems, roots, and shoots. Based on these observations, the culture medium influences the development of specific areas in the callus structure. Each of these areas may contain pluripotent and totipotent cells, and their specific arrangement inside the callus can further support organogenetic processes, including de novo shoot formation. The addition of cysteine to the culture medium in the pre-acclimation phase can further support the success of de novo shoot development for acclimation. Moreover, lessons learned from all experiments performed on *Syngonium*—including the principles applied to implement industrial-scale micropropagation—may further support the production of phenolic compounds relevant for in vitro systems at the industrial scale.

**Author Contributions:** Conceptualization, M.-M.A. and C.S.S.; methodology, M.-M.A.; validation, M.-M.A. and C.S.S.; formal analysis, M.-M.A.; investigation, M.-M.A.; resources, M.-M.A.; data curation, M.-M.A.; writing—original draft preparation, M.-M.A.; writing—review and editing, M.-M.A.; visualization, C.S.S.; supervision, C.S.S. All authors have read and agreed to the published version of the manuscript.

**Funding:** This research received no external funding.

**Data Availability Statement:** Not applicable.

**Acknowledgments:** This paper is dedicated to the memory of Aurelia Brezeanu, the mentor in the formation of Maria-Mihaela Antofie, who provided much counsel and inspiration to us both, for her pioneering work, valuable contributions, and untiring efforts in developing the science of plant cell, tissue, and organ cultures in Romania from 1970 to 2022.

**Conflicts of Interest:** The authors declare no conflict of interest.

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
