# Peer review of "De Novo Shoot Development of Tropical Plants: New Insights for Syngonium podophyllum Schott."

_horticulturae, doi:10.3390/horticulturae8121105_

Round 1

Reviewer 1 Report

The work submitted addresses a very interesting topic on the propagation of one of the most economically important ornamental plants: Syngonium podophyllum. This species, moreover, could have certain interests in the plants’ extracts industry due to its richness in specialized metabolites. The authors made a remarkable effort commenting the revised literature, and it is noticed the expertise that they have in the field. However, in my opinion, this review work does not match the needs of a Q1 Journal, as there are many major points to be addressed in order to develop a comprehensive review article. The specific comments on the introduction, methods, results, and discussion are commented in the attached doc. Finally, some minor comments, such as typos, and issues of style/format are observed. All these suggestions are also commented below, with the hope that they could serve to amend these important issues of this work before considering the resubmission.

Author Response

Dear respected Reviewer

We do appreciate the entire effort to read and review our article. We consider as precious all remarks and suggestions and therefore we tried to answer to all of them.

Sincerely

Authors 

Reviewer 2 Report

In this manuscript of review from Camelia Sava Sand and Maria-Mihaela Antofie, the authors summarized the publications from their laboratory and recent achievements in syngonium and other species from callus to de novo shoots regeneration. They discussed that the culture medium composition such as phytohormone, mineral, vitamin, and cysteine can affect the sygnonium callus induction and shoot regeneration. Especially, the authors discussed the xylematic-like structures (XLS) formation which is important for morphogenetic process in shoot regeneration from callus greatly depends on the mineral composition of medium through reviewing experimental results conducted on sygnonium in authors’ laboratory between 1996 and 2004.

This manuscript is comprehensively described previous work published in authors’ lab and recent advances in sygnonium tissue culture and regeneration. However, the manuscript is extremely difficult to read, and I wonder how many readers will undertake the effort to follow the authors. The authors use lots of long sentences with many grammatical mistakes which were hard to follow.  Here are additional comments:

1.       Line 183: “(Dianthys caryophyllus)” the annotation should go after when the word “carnation” first appears.

2.       Line 172-185: In this paragraph, the authors mainly discuss the first principle. But in Line 178, the authors begin to discuss the second principle. However, in the following paragraphs, they talk about the second principle with the same words again. I suggest that the authors should re-organize the two paragraphs.

3.       Line 220-221, re-organize this sentence. It is full of grammatical mistakes.

4.       The abbreviation “MS62” should be annotated when it first appears in line 223 rather than in line 393.

5.       Line226: “The success was 100% of starting meristem culture”. This simple sentence is unreadable. Does it mean that “the rate of success was 100% for the starting meristem culture”?

6.       Line 241: Asa consequence?

7.       Line 270-272: this extremely long sentence is full of mistake and unreadable. Please re-organize and shorten it.

8.       Line288-290: grammatical mistake. It is difficult to follow.

9.       Line 292: “the level of 1998 it was considered as profitable”?

10.   Line 300: “be used” should be “use”.

11.   Line133: “1,84” should be “1.84”.

12.   Line339-340: grammatical mistake, unreadable.

13.   Line 342: what’s the “TCP”? The abbreviation should be annotated.

14.   Line 563: “lot” should be “lots”.

Author Response

(The authors gave the same response as above.)

Reviewer 3 Report

The manuscript entitled: De novo shoots development of tropical plants – new insights for Syngonium podophyllum Schott, is well written and describes the methods of obtaining de novo shoots, also analyzing the histological aspect and phytohormonal regulation.

Major concerns:

However, despite the high value of the work, it is not coherent and clear. The reader cannot refer to all literature items, so there is no possibility to trace the methods used by the authors of the work.

Authors should revise the Abstract to be more compatible with the manuscript text.

Some of the information presented by the authors in the Introduction chapter is information at a basic level of knowledge and does not contribute anything to the work.

The author, referring to earlier publications, which are not all available to the reader, did not include e.g. information on the method of sterilization of plant material for the initiation of meristem culture.

What were the ingredients of the medium into which shoots were translated in subsequent passages?

After what time from the inoculation on the medium, the observation and calculation of the number of shoots was carried out?

The quality of the photos presented is very poor, the authors should improve it. There is also no reference scale for histological preparations. All the figure resolution should be improved.

Minor points:

Line 795, 825, 989 - the author has to correct errors in the text editing, they do not comply with the Editorial Board's guidelines.

The References do not comply with the Editorial Board's guidelines.

The author should correct minor errors such as: no spaces, e.g. line 314, 379

 Conclusion

The manuscript shows that the authors have a great deal of knowledge on the topic they describe. The work is very extensive and after some corrections and changes can be improved and published.

Author Response

Dear Professor,

It is important for us to recognize that the quality of this article is much better compared to the very first form. Therefore, all your recommendations and suggestions were further analyzed and tried to respond. We do appreciate all your comments.

Thank you

Authors

The manuscript entitled: De novo shoots development of tropical plants – new insights for Syngonium podophyllum Schott, is well written and describes the methods of obtaining de novo shoots, also analyzing the histological aspect and phytohormonal regulation.

Major concerns:

However, despite the high value of the work, it is not coherent and clear. The reader cannot refer to all literature items, so there is no possibility to trace the methods used by the authors of the work.

Authors should revise the Abstract to be more compatible with the manuscript text.

A: we arranged the introduction according to the Results and Discussion section and hope that we gave more compatibility.

Some of the information presented by the authors in the Introduction chapter is information at a basic level of knowledge and does not contribute anything to the work.

We introduced a text response towards this relevant point into the text line 132

The author, referring to earlier publications, which are not all available to the reader, did not include e.g. information on the method of sterilization of plant material for the initiation of meristem culture.

We introduced an appropriate text in 3.2.

What were the ingredients of the medium into which shoots were translated in subsequent passages?

We introduced an appropriate text in 3.2.

After what time from the inoculation on the medium, the observation and calculation of the number of shoots was carried out?

We introduced an appropriate text in 3.2.

The quality of the photos presented is very poor, the authors should improve it. There is also no reference scale for histological preparations.

 All the figure resolution should be improved.

We introduced the scale for each of the photos. And we may provide the original photos

Minor points:

Line 795, 825, 989 - the author has to correct errors in the text editing, they do not comply with the Editorial Board's guidelines.

A: complied to the suggestions

The References do not comply with the Editorial Board's guidelines.

A: we tried our best to re-verify some of the references changed

The author should correct minor errors such as: no spaces, e.g. line 314, 379

A: we tried to correct these errors

 Conclusion

The manuscript shows that the authors have a great deal of knowledge on the topic they describe. The work is very extensive and after some corrections and changes can be improved and published.

Round 2

Reviewer 1 Report

Dear authors, 

Thanks for the revised paper, and the comments to the previous revision. I acknowledge the effort in including more information, and improving the manuscript. However, there are still major issues to be ammended as they are essential aspects of a good review paper. 

- According to the methodology followed, the topic does not gain a good coverage, as there is no extensive (and critical) review of the available information published on in vitro cultures of Syngonium podophyllum. Instead, there are only covered few publications on the species (and easily accessible through interenet, not all). A simple search in WoS or Google Scholar can provide you a much higher number of references (however, I can't repeat the search as it is done because it is not sufficiently explained in MyM).

- In my opinion, to select (for review) only the papers published by the lab is not enough reason to publish a review in a Q1 Journal such as Horticulturae.The justification made by the authors not scientifically based, and therefore, I would recommend to submit an ammended version to a divulgative Journal (as the result is to illustrate the expertise of the group on the topic, rather than making a comprehensive and criticall coverage of the topic in order to reach a wide audience of readers).

- In line with the previous observation, Methodology is still not sufficiently developed. The authors referred other articles to check the methodology, but (in my humble opinion) a good review must be precise in explaining the search strategy in a clear way (including period of coverage, keywords, inclusion, and exclusion criteria, etc.). 

- None of the authorities of plant species was included, as was requested in the previous version (and also, authors said that they checked using ipni). Examples are in lines 63, 225, 336, 587, 597, 600, and many more.

-  Also, there are errors when they mention the cultivar, they express the results in a very confusing way (they do not provide concentrations when explaining the hormone balance), among other format errors such as italics in places where are not needed (lines 268, 504), lack of italics (line 290), double spaces (315, 332, 379, and some more), repetitive writing (lines 141-143; 276-279). Also, species names can be abbreviated after the first mention in the text, but must be in the complete form in the illustrations (illustrations must be self-explaining). 

- It is still difficult to follow, maybe due to the fact that the text was not revised by a native. Therefore, an extensive edition of the English language is also needed, as I proposed in the first revision.

I expect all these suggestions serve for further versions of the manuscript. 

Author Response

Dear Professor,

It is important for us to recognize that the quality of this article is much better compared to the very first form. Therefore, all your recommendations and suggestions were further analyzed and tried to respond. We do appreciate all your comments.

Thank you

Authors

Dear authors, 

Thanks for the revised paper, and the comments to the previous revision. I acknowledge the effort in including more information, and improving the manuscript. However, there are still major issues to be ammended as they are essential aspects of a good review paper. 

- According to the methodology followed, the topic does not gain a good coverage, as there is no extensive (and critical) review of the available information published on in vitro cultures of Syngonium podophyllum. Instead, there are only covered few publications on the species (and easily accessible through interenet, not all). A simple search in WoS or Google Scholar can provide you a much higher number of references (however, I can't repeat the search as it is done because it is not sufficiently explained in MyM).

A: we added at least other 8 references.

- In my opinion, to select (for review) only the papers published by the lab is not enough reason to publish a review in a Q1 Journal such as Horticulturae. The justification made by the authors not scientifically based, and therefore, I would recommend to submit an ammended version to a divulgative Journal (as the result is to illustrate the expertise of the group on the topic, rather than making a comprehensive and criticall coverage of the topic in order to reach a wide audience of readers).

- In line with the previous observation, Methodology is still not sufficiently developed. The authors referred other articles to check the methodology, but (in my humble opinion) a good review must be precise in explaining the search strategy in a clear way (including period of coverage, keywords, inclusion, and exclusion criteria, etc.). 

A: Thank you for this precise suggestion. We developed further hope accordingly

- None of the authorities of plant species was included, as was requested in the previous version (and also, authors said that they checked using ipni). Examples are in lines 63, 225, 336, 587, 597, 600, and many more.

A: We think that there is a confusion between our texts as we cannot identify by rows. We introduced one of the suggested link

-  Also, there are errors when they mention the cultivar, they express the results in a very confusing way (they do not provide concentrations when explaining the hormone balance), among other format errors such as italics in places where are not needed (lines 268, 504), lack of italics (line 290), double spaces (315, 332, 379, and some more), repetitive writing (lines 141-143; 276-279). Also, species names can be abbreviated after the first mention in the text, but must be in the complete form in the illustrations (illustrations must be self-explaining). 

A: we tried to go through all the document. It is a bit hard due to track changes

- It is still difficult to follow, maybe due to the fact that the text was not revised by a native. Therefore, an extensive edition of the English language is also needed, as I proposed in the first revision.

A: We will go for an EN revision of the text if it will be accepted before publication

I expect all these suggestions serve for further versions of the manuscript. 

Round 3

Reviewer 1 Report

Dear authors, please, be sure that all authorities are mentioned, and all typos are corrected before its publication. Also, it is of capital importance to check carefully the English. 

Yours sincerely.

Author Response

Dear authors, please, be sure that all authorities are mentioned, and all typos are corrected before its publication. Also, it is of capital importance to check carefully the English. 

A:  Dear professor

We do appreciate your immense effort for supporting us developing our research study. We add another authority, corrected the scientific names and contacted another colleague to read the EN language